# Particle Size Effect of Oyster Shell on Mortar: Experimental Investigation and Modeling

**DOI:** 10.3390/ma14226813

**Published:** 2021-11-11

**Authors:** Yingdi Liao, Hongyi Shi, Shimin Zhang, Bo Da, Da Chen

**Affiliations:** 1College of Harbour, Coastal and Offshore Engineering, Hohai University, Nanjing 210098, China; liaoyingdi@hhu.edu.cn (Y.L.); 211303020030@hhu.edu.cn (H.S.); ximichang976@163.com (S.Z.); dabo@hhu.edu.cn (B.D.); 2Key Laboratory of Coastal Disaster and Defence of Ministry of Education, Hohai University, Nanjing 210098, China; 3Yangtze Institute for Conservation and Development, Hohai University, Nanjing 210098, China

**Keywords:** waste oyster shell, mortar, particle size, compressive strength, flexural strength, static elastic modulus, dry shrinkage

## Abstract

In order to solve the problem of lack of natural river sand, crushed waste oyster shells (WOS) were used to replace river sand. By replacing 20% river sand, WOS mortar with different particle sizes of WOS were made for the experiment. Through experimental observation, the initial slump and slump flow loss rate were studied. The effects of different particle sizes and curing times on the compressive strength, flexural strength, static elastic modulus, and dry shrinkage of WOS mortar were analyzed. The relationship formulas between the compressive strength, flexural strength, particle size, and curing age were proposed. The results showed that the setting time and slump flow decreased with a decrease in the particle size of WOS. It was also found that the mortar with fine crushed WOS had high compressive strength, flexural strength, and static elastic modulus at both early and long-term curing age. A formula was proposed to describe the development of the compressive strength with the particle size of WOS and curing time, and the relations among these mechanical properties were discussed. Furthermore, drying shrinkage increased when WOS was used and could not satisfy the standard requirement of 0.075%. In contrast, the addition of fine WOS and double-dose sulfonated naphthalene-formaldehyde superplasticizer (SNF SP) reduced the shrinkage rate of the mortar by 8.35% and provided better workability and mechanical properties for mortar.

## 1. Introduction

Since aquaculture is one of the most critical components of businesses in coastal countries, the disposal of a large number of abandoned oyster shells appears to be a top priority for many countries [1,2,3]. Similar to many other coastal countries such as Japan, South Korea, etc., China is rich in oyster shells. Along the east coast of China, over 5 million tons of oyster shells were generated in 2010, and the amount of shell production increased year by year quickly. Meanwhile, the total output of concrete in China was 2.9 billion cubic meters, of which the output of concrete in coastal engineering was about 800 million cubic meters in 2020 [4], and fine aggregate occupies about 160 million cubic meters. Recently, researchers have tried to recycle the waste oyster shells (WOS) as a constituent of construction materials [5,6]. Oyster shells are mainly composed of CaCO_3_ (approximately 96%), which has no adverse effect on the cement hydration and only plays a role as the filler in cement matrix [7,8]. The WOS is usually crushed into small particles and used as a substitute for fine aggregate. Due to the excessive flakiness of particles, replacing coarse aggregate with the whole shell appears to be less effective, which will result in large voids in the sample [9]. In addition, economic and environmental restrictions have harmed the supply of natural sand. WOS, as a partial replacement of fine aggregate, can alleviate the mining of natural sand and satisfy the growing demand for natural sand.

Early researchers such as Wang [10] and Yoon [11] concluded that the density, compressive strength, and static elasticity modulus of the specimens decreased with the workability increase. It was caused by the use of WOS instead of the river sand in mortar. Cuadrado-Rica [12] observed that the replacement of crushed shell as aggregates would increase the air entrainment in the concrete, and the workability and mechanical properties of the specimens would be lower than that of ordinary concrete. In his work, when the fine aggregate of concrete was replaced by 20%, the compressive strength of the specimen decreased by around 9.9% at 28 days. However, the research of Safi [13] showed that the mixed oyster shell would not significantly influence the specimen in compressive strength because of the excellent adhesion between shell and cement. WOS is acknowledged as suitable for use in cement-based materials, and a maximum of 40% replacement is suggested in the report [7]. Conclusions drawn from the reported studies are quite uncertain owing to different experiment parameters. One of the critical influencing factors is identified as the particle size of aggregate. Lertwattanaruk [14] studied four kinds of mortars applied with crushed seashells of different species and found that the mortars with good strength development had a common characteristic of containing small shells. Furthermore, previous studies pointed out that the fine particle proportion of aggregate considerably influences the workability and mechanical properties of the cement-based material. The research conducted by Yang [15] was a pioneer study of the particle size effect of the oyster shell in concrete. The fineness modulus (F.M.) of the two kinds of WOS used in his work for 10% sand replacements were 2.1 and 2.7, respectively. The experiment results showed that concrete with the finer WOS was 13.6% higher in compressive strength than the control group at 28 days, while the other one was 6.2% lower. In Dang and Boutouils’ report [16], crushed seashell was used as a 20% aggregate replacement in making pervious concrete pavers. The addition of seashell ranging around 2–4 mm decreased the porosity of concrete and increased its mechanical strength in contrast to the addition of 4–6.3 mm seashell. Benabed [17] revealed that adding a suitable dose of fine particle could improve the concrete performance in compressive and flexural strength while weakening its permeability. Since the specific shape and configuration of the aggregation result in a redistribution of pores in the concrete, more fine particle help produce a more compact product couple with a higher magnitude of capillary pores, promoting the water infiltration [18,19]. It is no doubt that the influence of the particle size of WOS on the properties of cement-based materials is crucial.

From the above, there are indeed a lot of research studies on the WOS-based material. However, most of them focused on the effect of the proportion and the species of the shell, while very few studied the effect of their particle size. Hence, there remains a need to carry out further studies. The study aims to give systematical research about the effect of particle size of WOS on the fresh and hardened properties of mortar. Crushed coarse, middle, and fine WOS were used as 20% fine aggregate replacement in mortar. A series of properties, including slump flow, setting time at the fresh stage, flexural strength, compressive strength, static elasticity modulus, water absorption, and drying shrinkage at the hardening stage were tested. Test results were analyzed and discussed, leading to a conclusion of a rational design for reusing the waste oyster shell in construction materials.

## 2. Experimental

### 2.1. Materials

#### 2.1.1. Cement

Ordinary Portland cement (OPC) of strength class 42.5 R conforming to ASTM Type Ⅰ was used as binding material [20]. The cement was obtained from Conch China Cement Co., Ltd. in Nanjing, China. The specific gravity and the fineness of OPC were 3.16 g/cm^3^ and 3519 cm^2^/g. The chemical compositions and mineral compositions of the OPC are listed in Table 1, which were provided by the factory. The performance indexes of cement meet the standard of GB 175 (2007) requirements.

#### 2.1.2. Fine Aggregates

Natural river sand (siliceous sand) and waste oyster shells (WOS) of different particle sizes were used as fine aggregates to prepare mortar. All the sand and WOS have a maximum diameter of 5 mm. Figure 1 presents the WOS, which seems flakier and more elongated than the ‘cubical’ shape of sand. The crushed WOS was sourced from the southeast coast of China. WOS were first rinsed to remove contaminants. Then, they were dried at 105 °C in a drying oven for 5 h and crushed using a crushing machine. Finally, WOS with coarse, medium, and fine particle sizes were obtained. Figure 2 presents the particle size distribution curves of WOS compared with sand specified by ASTM C1585 (2013). All (100%) of natural river sand passed through a 4.75 mm sieve; 90% of the coarse particles have a size around 1.18–4.75 mm with a median particle size of 2.71 mm, while 90% of the fine particles have a size around 0–1.18 mm with a median particle size of 0.56 mm, and the median particle sizes of the middle particle is 1.45. The density of the coarse, medium, and fine crushed WOS particles is 1284 kg/m^3^, 1299 kg/m^3^, and 1354 kg/m^3^, respectively. The water absorptions of the coarse, middle, and fine WOS are 6.03%, 8.52%, and 11.85%, and fineness modulus (F.M.) values of the coarse, middle, and fine are 2.37, 3.48, and 4.51, respectively. The F.M. of the natural river sand is 1.59.

#### 2.1.3. Superplasticizer (SP)

In this study, a sulfonated naphthalene-formaldehyde (SNF) superplasticizer (SP) with 99% purity was utilized to facilitate the workability of the fresh mortar. It was provided by Jiangsu Bote New Materials Company in Nanjing, China. According to the pre-test results, the amount of water-reducing agent for the premise of meeting the workability requirements for each WOS mortar was different, but the amounts were all close to 0.25%. To compare the properties of WOS mortars with different particle sizes, the same amount of SNF SP (0.25%) was used.

### 2.2. Mixture Proportions

To investigate the effect of the particle size of WOS on properties of fresh and hardened mortar, three kinds of mortars (WOS-C, WOS-M, WOS-F) were prepared with river sand, 20% of which had been replaced by WOS of different grades (20% is the most widely used ratio in previous studies). The porosity of WOS mortar with different particle sizes (WOS-F, WOS-M, and WOS-C) is 8.74%, 10.26%, and 10.48%, respectively. The recipe for the test mortar is listed in Table 2. All the mixtures have the same mix proportion, and the cement content, water-to-cement ratio (W/C), and amount of fine aggregate was kept at 608 kg/m^3^, 0.45, and 1520 kg/m^3^, respectively. The fine aggregate was kept at a fixed weight ratio of 2.5 to OPC. The SNF SP used in the mixture was kept at 1% by weight of OPC. According to Table 2, the range of global F.M. is 1.75–2.18, all of which belong to silver sand of 1.65–2.2.

### 2.3. Sample Preparation and Curing Conditions

Ingredients of the mortar mixture were fine aggregate (natural river sand and crushed WOS), OPC, and SNF SP. The aggregates used here were in saturated surface dry condition. (First, the aggregate is soaked in water to reach a certain humidity. Then, the aggregate was taken out and dried naturally in the laboratory until there was no obvious moisture on the aggregate surface. All the internal pores of aggregate are saturated with water absorption, but there is no obvious water on the surface.) Firstly, all of those were batched by weight. River sand, crushed WOS, and OPC were premixed together for 3 min using a standard laboratory rotary mixer. Then, the water containing SNF SP was added into the mixture, and the mixing continued for a total period of 5 min. Fresh mortar was filled into the standard steel molds for corresponding tests. All the molds were covered by polyethylene films at the surface to prevent from moisture loss and were cured at a temperature of 20 ± 3 °C. Finally, samples were demolded after 24 h and cured in a standard condition with a temperature of 20 ± 3 °C and a relative humidity of 90 ± 5%. Three mortar samples were prepared for each test in this study.

### 2.4. Properties of Fresh Mortar

Immediately after the mixing process, fresh mortar was prepared for the slump flow test according to EN 1015-3 (1999) [21]. The slump flow loss rate was measured by the variations of the slump flow with elapsed time. The setting time test was conducted based on EN 480-2 (2006) [22] with a Vicat needle. The initial setting time is identified as the time when cement paste starts losing its plasticity, while the final setting time corresponds to the time when the paste has completely lost it plastic property. The setting time was determined by observing the needle penetrating into mortar until it reached a specified value. The average of three measurements was the value of the setting time.

### 2.5. Properties of Hardened Mortar

#### 2.5.1. Mechanical Property

According to EN 1015-11 (1999) [23], prism samples with a size of 40 mm × 40 mm × 160 mm were tested for compressive strength and flexural strength at 3, 7, 28, and 90 days. Flexural strength was tested through three-point loading experiments. The broken samples obtained here were reused for compressive strength measurements. The compressive strength and flexural strength were the average of three measurements, respectively.

#### 2.5.2. Static Elasticity Modulus

Figure 3 presents the static elasticity modulus experiments. The static elasticity modulus is tested according to the standard of EN-197-1-211. According to the specification, the cylinder test block is made of steel mold. The static elasticity modulus test was carried out on a cylinder sample with a diameter of 50 mm and a height of 100 mm using an auto-compensated and auto-equilibrated system of TOP INDUSTRY, Franc. An axial stress generator with a capacity of 375 MPa, and two linear variable displacement transducers (LVDT) fixed at both two opposite sides of the sample were used in this system. The test method of the static modulus is as follows: The relationship between deformation and stress is obtained through an equal strain loading method. Then, taking two points in the elastic segment to calculate the slope, the static elasticity modulus is achieved in the end.

#### 2.5.3. Water Absorption and Drying Shrinkage

The water absorption of the mortar was tested at 28 days according to ASTM C1585 (2013) [24]. This test determined the sorptivity of the mortar by measuring the increase in the mass of water absorbed by a sample as a function of time. The drying shrinkage of the mortar was tested according to ASTM C157/157M (2008) [25]. Samples were cured in an environment with a temperature of 23 °C and relative humidity of 50%; then, they were tested at 1, 3, 7, 14, 28, 56, 90, and 112 days.

#### 2.5.4. Stress–Strain Curves

Three cylindrical samples of φ 50 mm × 100 mm were used for the stress–strain test. Before the stress–strain of the sample was measured, a linear variable displacement transducer (LVDT) was fixed on two opposite sides of the sample. The test of uni-axial compression was carried out at a constant strain rate of 0.001 mm/s using the auto-compensated and auto-equilibrated triaxial cell system.

## 3. Results and Discussion

### 3.1. Properties of Fresh Mortar

#### 3.1.1. Initial Slump Flow and Slump Flow Loss Rate

Figure 4 presents the development of slump flow with time elapse for all the mortars. It can be noted that the particle size of waste oyster shell (WOS) has a significant influence on the slump flow. For samples casting with crushed fine, middle, and coarse WOS, the initial slump flows were 116 mm, 127 mm, and 140 mm, respectively. WOS-F and WOS-M showed reductions of initial slump flow of 17.14% and 9.28% compared to WOS-C. The initial slump flow of the mortar was found to increase with the increase in the particle size of aggregate, which is consistent with the existing research conclusions [26]. The drop of initial slump flow caused by using WOS with a smaller particle size is mainly due to the larger antiparticle friction resulting from the more irregular surface and larger specific surface area of the particle. Meanwhile, WOS-F contains more fine particles, which would absorb more water for the mortar mixing, thus decreasing the free water in the sample and reducing the working performance [27].

Moreover, Figure 4 shows that the slump flow loss of WOS-C was 22.14% during the 2 h testing time; correspondingly, higher slump flow losses of 28.85% and 29.43% were found in WOS-M and WOS-F. The small slump flow loss of the mortar with WOS of large particle size indicates that adding the coarse particle contributed to the more stable workability of the mortar.

#### 3.1.2. Setting Time

Figure 5 presents that both the initial and final setting times increase with an increase in particle size. The initial setting times of WOS-C, WOS-M, and WOS-F were 516 min, 492 min, and 464 min, respectively; all of them were able to meet the EN 197 (2011) [28] standard requirement (≥60 min). The test result showed that WOS-C has a 52 min longer initial setting time and a 68 min longer final setting time than WOS-F. The large particle size of WOS prolonged both the initial and final setting time. Lu [29] explained that the aggregate of large particle size resulted in the remarkable maintaining of free water available for the hydrolysis, which contributes to an increase in the effective water-to-cement ratio. The cement paste with a higher water-to-cement ratio is known as having better workability and taking a longer time to form a rigid structure. Furthermore, the initial lower water absorption of coarse WOS mentioned previously forcefully supports this explanation, since less water is consumed in mixing process, and more free water exists in the paste.

### 3.2. Properties of Hardened Mortar

#### 3.2.1. Compressive Strength

Figure 6 presents the result of the compressive strength test. The compressive strengths of all the samples grew with the increase in the curing period and the decrease in the particle sizes of WOS. At 3 days, the compressive strengths of WOS-C, WOS-M, and WOS-F were 21.21 MPa, 22.15 MPa, and 22.98 MPa, respectively. Then, the strengths increasing by 64.68%, 64.97%, and 63.7% were found at 28 days compared with 3 days, while merely rising by 7.74%, 6.79%, and 7.07% at 90 days compared with 28 days. The compressive strength of the mortar rose rapidly during the first 28 days, and the progress slowed down at the further curing period. This is attributed to the gradual completion of the cement hydration reaction process [30]. It is shown that the smaller the porosity of the WOS, the denser the WOS mortar, and the greater the compressive strength and flexural strength of the WOS mortar, which are consistent with previous research conclusions [30].

Investigation showed that the compressive strength development of the mortar can be formulated according to ACI 209 as follows [31,32]:(1)$$f_{c}\left(t\right)=\frac{t}{A+Bt}f_{c}^{\prime }$$
where *f*_c_(*t*) is the compressive strength according to age (in days), and *f*_c_’ is the compressive strength at 3 days. The parameters A and B are strongly related to the strength development rate at early age and long-term age, respectively. It is proved that the higher the A value, the lower *f*_c_(*t*) will be at early age. Meanwhile, the lower the B value, the higher *f*_c_(*t*) will be at long-term age [33]. Figure 5 also presents the fitting curves and formulas for parameters A and B of each mortar obtained from nonlinear multiple regression analysis.

The compressive strengths of WOS-C, WOS-M, and WOS-F were in the ranges of 21.21–37.63 MPa, 22.15–39.02 MPa, and 22.98–40.28 MPa during the test 90 days. It is noted that WOS-F had the highest compressive strength during the whole age. Considering the fineness modulus (F.M.) to be a parameter to predict A and B, the following equations can be derived from this study:(2)$$A=1.22+0.03f_{m}$$
(3)$$B=0.535+0.0057f_{m}$$
where *f*_m_ is the fineness modulus of WOS.

Figure 7 presents the correlations between the fitting parameters and the F.M. of WOS. It clearly indicates that both parameters increase with the F.M. According to Hamza [34], the mechanical properties of cement-based material appeared related to the pore structure of the fragile biomaterial used as the bone substitute. WOS-F with the finest pores and the most compact particle heap resulted in the highest compressive strength. Sevim and Roy [35,36] explained that with the fine particle participating, the mixture components are in close contact and form a dense structure that results in the high compressive strength of the mortar.

After combing Equations (1)–(3), the compressive strength of the mortar could be finally determined by particle size and curing age:(4)$$f_{c}\left(t\right)=\frac{t}{0.005f_{m}t+0.03f_{m}+0.535t+1.22}f_{c}^{\prime }$$

#### 3.2.2. Flexural Strength

Figure 8 describes how the development of flexural strength of the mortar varies with curing age and particle size. It can be observed that the flexural strength grew fast at an early age and turned stable gradually, which was similar to the compressive strength. Researchers are always engaged in establishing a relationship between the two strengths. The smaller the porosity, the denser the mortar, and the greater the flexural strength will be, which are in line with previous research conclusions [30].

Figure 9 describes the flexural strength and compressive strength, which show a strong linear correlation, with an R-square value of 0.986:(5)$$f_{f}\left(t\right)=0.114f_{c}\left(t\right)+4.57$$
where *f*_f_(*t*) is the flexural strength according to age (in days).

The result is well corroborated by Latroch and Coppola [37,38]. With the equation, the compressive strength and flexural strength of WOS-based mortar can be mutually predicted.

Figure 8 also presents that during the test time, WOS-F with finer particles showed a higher flexural strength than WOS-M and WOS-C. In existing studies, factors such as the shape, size, volume fraction, and surface texture of the aggregate were proved to have an influence on the fracture property of hardened mortar [39,40]. Tasdemir [41] claimed that the cement–aggregate interface is one of the most fragile places. Hence, cracks usually develop at this interface, and the brittleness increases significantly, especially for mortar with coarse aggregate. Another fragile place is reported as the texture of the material. Since the surface of WOS is textured with indentations arranged along parallel rows, cracks usually occur on this weak plane and develop gradually [5]. Consequently, the coarse WOS is prone to large cracks and decrease in the flexural strength of sample. For mortar with different WOS, there always exists an equation to depict the relationship between flexural strength and compressive strength.

#### 3.2.3. Static Elasticity Modulus

Figure 10 shows that the transformation of the static elasticity modulus was seen to increase first and become stable later. This was similar to the compressive and flexural strength for the same reason that the cement hydration is gradually completed. From Figure 10, it also can be seen that the smaller the WOS particle size is, the smaller the porosity of the WOS mortar, the denser the WOS mortar, and the better its mechanical properties, making it stronger in resisting deformation under the same load. The smaller the WOS particle size is, the larger the elastic modulus of the WOS mortar, which are in line with previous research conclusions [42].
(6)$$E\left(t\right)=3.51\sqrt{f_{c}\left(t\right)}-3.9$$
where *E*(*t*) is the static elastic modulus according to age (in days).

Figure 11 presents the fitting curve with experimental data. Moreover, according to Section 3.2.2, the expression for the static elastic modulus of the mortar with flexural strength is given by combing Equations (5) and (6):(7)$$E\left(t\right)=10.4\sqrt{f_{c}\left(t\right)-4.57}-3.9$$

#### 3.2.4. Stress–Strain Curves

Figure 12 shows that WOS-F has the maximum stress and strength, while WOS-C has the minimum stress and strength, indicating that the smaller the particle size is, the greater the strength of the mortar. With the extension of curing age, the stress peak increases, which conforms to the law that the longer the curing age is, the higher the strength grade and the stronger the brittleness of the mortar, which is similar to ordinary concrete [43].

Figure 13 shows the stress–strain curves of WOS-F, WOS-M, and WOS-C after the stress value was normalized. It can be found that the descending segment of WOS-F was steeper than that of WOS-M and WOS-C, indicating that the brittleness of WOS-F was stronger than that of WOS-M and WOS-C. This was mainly because compared with WOS-M and WOS-C, the particle size of the WOS-F aggregate was smaller, the internal void was smaller, and the strength was larger, so the brittleness was larger. When reaching the peak stress, WOS-F was destroyed faster. It is shown that the smaller the WOS particle size is, the smaller the porosity of the WOS mortar, the denser the WOS mortar, the higher the strength, the steeper the descending section, and the greater the brittleness, which are consistent with previous research conclusions [44].

#### 3.2.5. Drying Shrinkage

Figure 14 presents the drying shrinkage of the mortar containing crushed WOS over time. It can be seen that the drying shrinkage increased rapidly in the first stage and slowed down in later stages. Mortar prepared with finer WOS has a larger drying shrinkage all the time. One source of the drying shrinkage is that the water held in the capillary pores of the hydrated cement paste runs away to the environment [44]. Finer particles result in a more compact structure, and a higher magnitude of capillary pores in the mortar helps create a larger shrinkage. A study conducted by Collins [45] about the relationship between the pore size and the drying shrinkage on the concrete also supported the result. Kuo [5] related the drying shrinkage with the water absorption of aggregate. He claimed that the aggregate with higher water absorption causes more shrinkage in cement-based material.

The dry shrinkage of the mortar is mainly determined by the capillary tension in the mortar [46]. In addition, WOS can effectively improve the particle size distribution and refine the pore structure in mortar. In addition, the finer the pore structure of mortar is, the smaller the internal capillary tension. With the same amount of WOS, the smaller the aggregate particle size is, the smaller the pore size of the mortar and the more refined the pore structure of the mortar. Therefore, the smaller the WOS particle size is, the smaller the dry shrinkage value, which is consistent with previous research conclusions [47].

It can be found that after curing by 90 days, the drying shrinkage of the WOS-F has exceeded 0.075%, which is beyond the value required by the standard according to AS 3600 (2004). Faced with such a problem, researchers prefer to add more water reducer into mortar to release the water entrapped in the cement clusters [48,49]. Hence, an additional set of mortar (WOS-SP) prepared with fine WOS and double-dose SNF SP and corresponding 85% water was supplemented in this study. As shown in the figure, the addition of more SNF SP restrained the shrinkage of the mortar significantly. WOS-SP has a decrease by 8.35% in shrinkage compared with WOS-F at 112 days. The WOS-SP tested has a better performance in workability and mechanical strength.

### 3.3. Eco-Efficiency

Oyster shells occupy land and marine resources, produce harmful substances, pollute the air and water, destroy the ecological environment, and waste resources. However, due to people’s weak awareness of environmental protection and lack of corresponding technical support, oyster shells are not correctly handled in China.

By using the crushed waste oyster shell (WOS) to replace 20% river sand, the technique in this paper greatly alleviates these problems. As a non-renewable resource, river sand is increasingly scarce. Replacing river sand with WOS could not only reduce the demand for river sand but also properly dispose of waste WOS. In a word, by recycling WOS as a river sand substitute in mortars, the sustainable recycling of WOS and cleaner production of eco-friendly and high-performance mortars can be realized. This environment-friendly engineering method is worth popularizing.

## 4. Conclusions

The analysis shows that crushed waste oyster shells (WOS) simply processed can partially replace fine aggregate in conventional mortar and provide good engineering properties. Based on the experimental results from mortars containing WOS of different particle sizes, the following conclusions can be drawn:

(1) The workability of WOS-based mortar was influenced by the particle size of aggregate significantly. Compared with mortar casted with fine WOS, one with coarse WOS presented a larger slump flow and a lower slump flow loss rate. However, the workability of all the WOS-based mortar was within the standard requirement.

(2) The maximum compressive strength, flexural strength, and static elasticity modulus of the hardened mortar were obtained through using the WOS of the small particle size, which approved that the fine WOS outperformed the coarse WOS in terms of producing a harder and more brittle mortar.

(3) The development of compressive strength of WOS-based mortar can be well described by the modified ACI function, which considers both the particle size of WOS and the curing time of specimens. Meanwhile, a linear relationship exists between compressive strength and flexural strength as well as between the static elasticity modulus and square root of compressive strength, regardless of the different aggregates.

(4) The drying shrinkage of the mortar containing fine WOS was higher than those of the mortar with coarse WOS, which cannot satisfy the standard requirement of 0.075%. The addition of more superplasticizer reduces the shrinkage rate of WOS-SP by 8.35% compared with WOS-F at 112 days, satisfying the standard requirement of 0.075% and providing a better workability and mechanical properties for mortar.

Considering the lack of the natural aggregate and environmental problem caused by the disposal of waste seashells, a cheap and eco-friendly WOS-based mortar with good workability and mechanism is proposed by using fine WOS.

## Figures and Tables

**Figure 1 materials-14-06813-f001:**
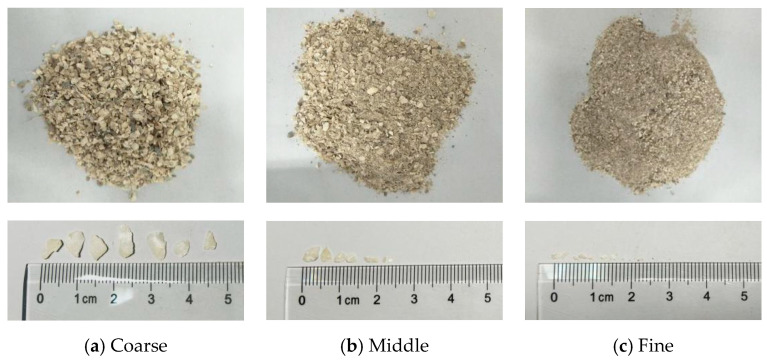
WOS of different particle sizes. (**a**) Coarse. (**b**) Middle. (**c**) Fine.

**Figure 2 materials-14-06813-f002:**
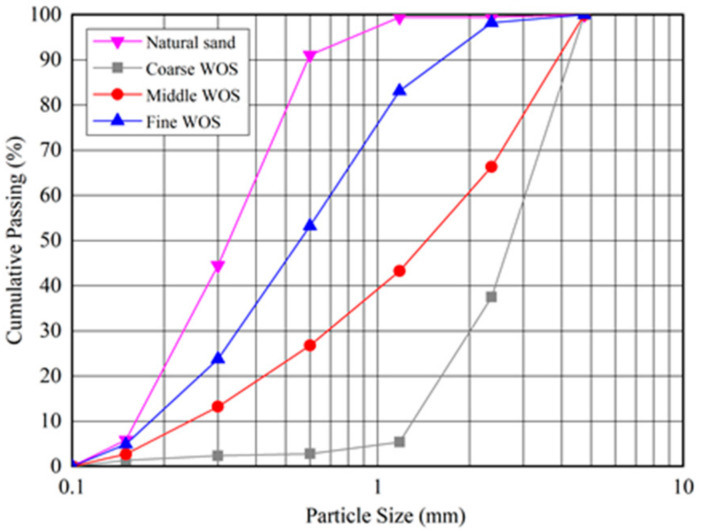
The particle size distributions of aggregates.

**Figure 3 materials-14-06813-f003:**
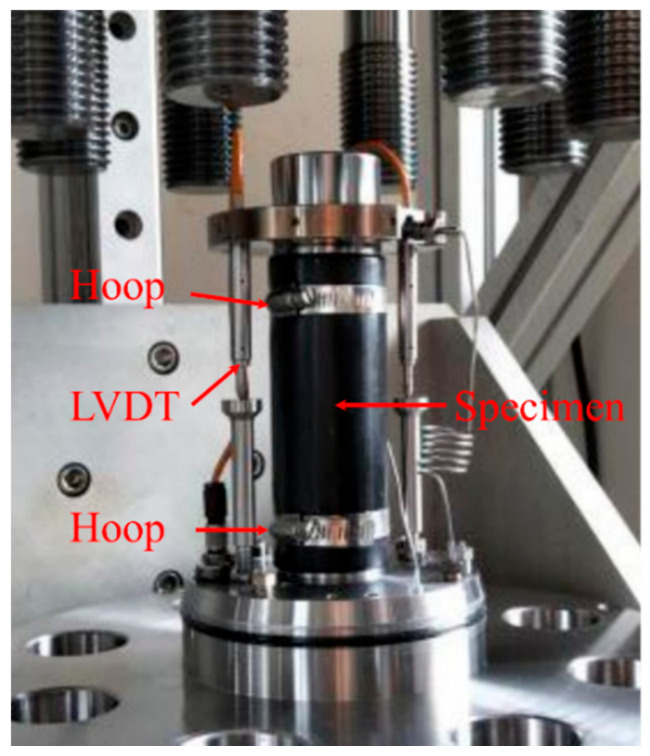
The static elasticity modulus experiments.

**Figure 4 materials-14-06813-f004:**
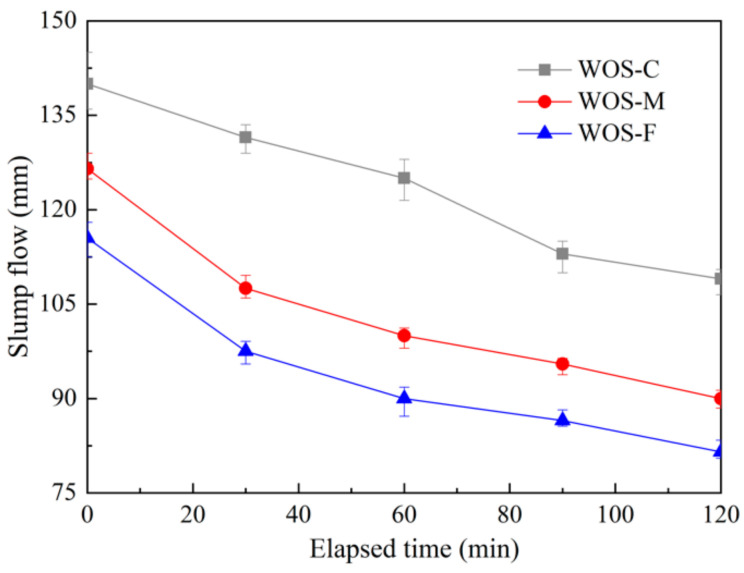
The effect of particle size of WOS on the initial slump flow.

**Figure 5 materials-14-06813-f005:**
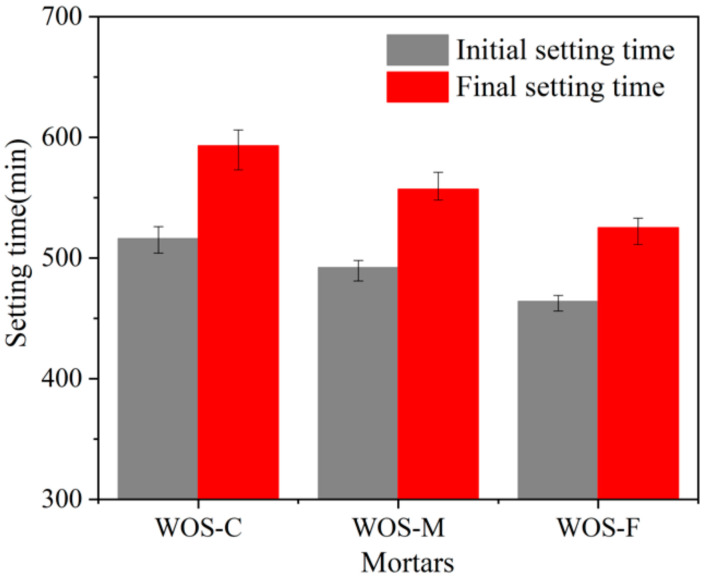
The effect of particle size of WOS on the setting time.

**Figure 6 materials-14-06813-f006:**
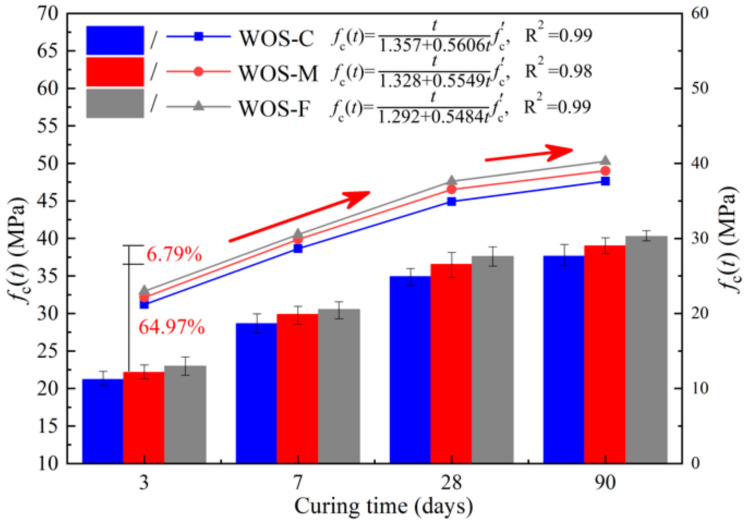
The effect of particle size of WOS on the *f*_c_(*t*).

**Figure 7 materials-14-06813-f007:**
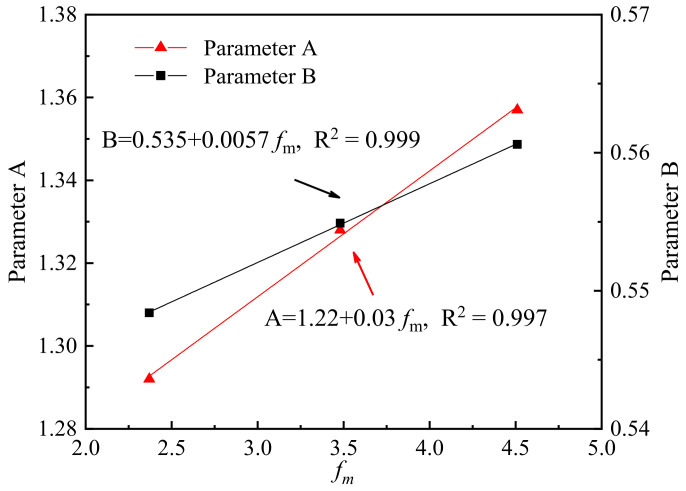
Correlations between the fitting parameters and the fm.

**Figure 8 materials-14-06813-f008:**
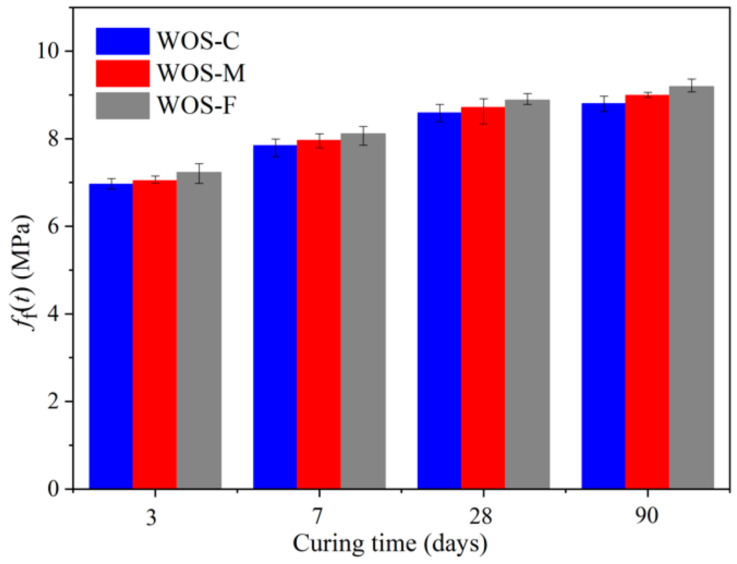
The effect of particle size on the *f*_f_(*t*).

**Figure 9 materials-14-06813-f009:**
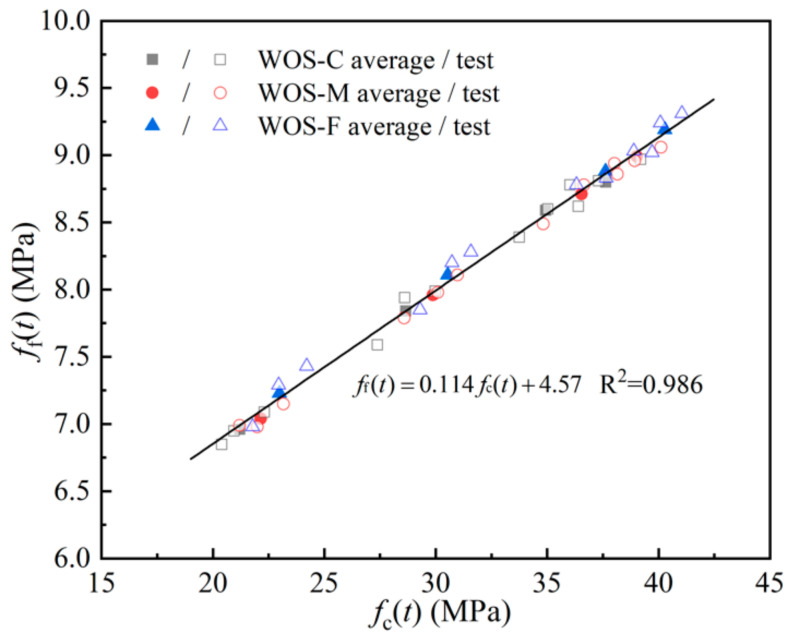
The correlation between *f*_f_(*t*) and *f*_c_(*t*).

**Figure 10 materials-14-06813-f010:**
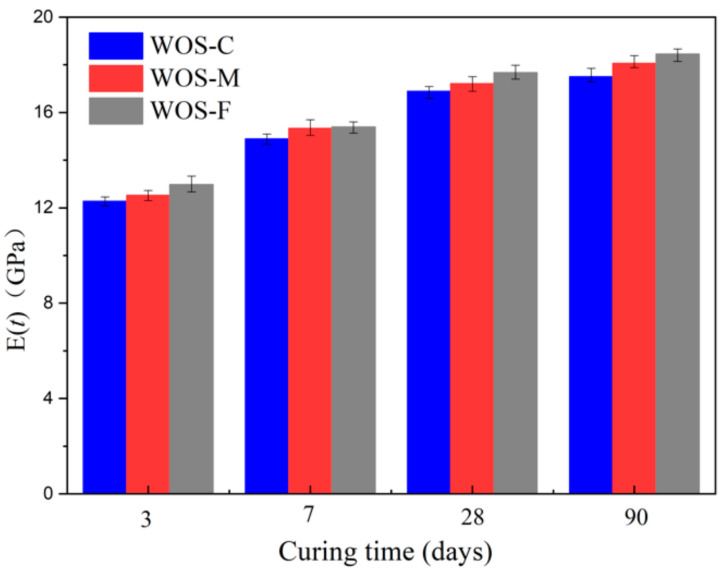
The effect of particle size on the static elastic modulus.

**Figure 11 materials-14-06813-f011:**
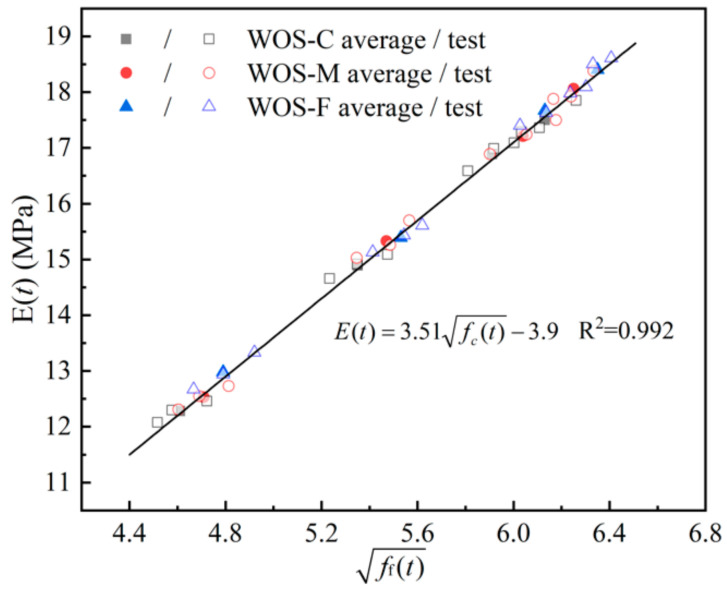
The correlation between *E*(*t*) and *f*_c_(*t*).

**Figure 12 materials-14-06813-f012:**
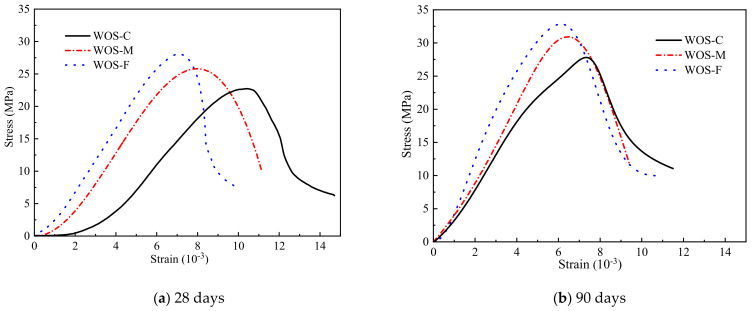
The effect of the particle size on the stress–strain. (**a**) 28 days. (**b**) 90 days.

**Figure 13 materials-14-06813-f013:**
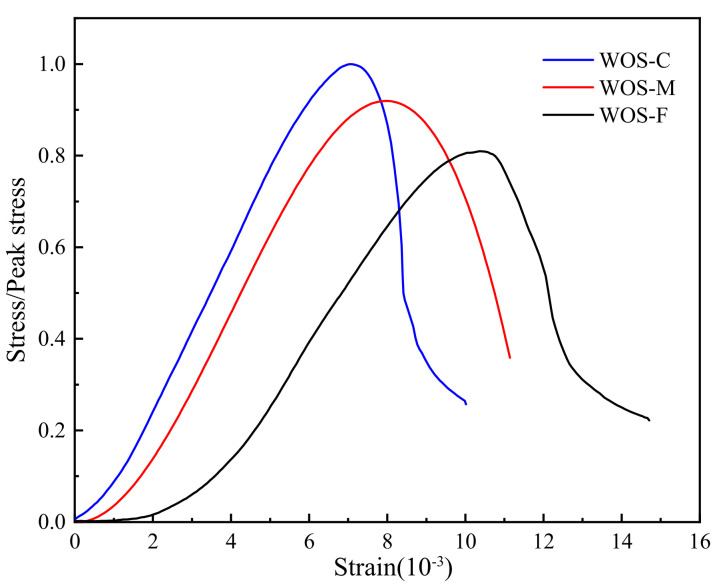
Normalized stress–strain curves of different particle size at 28 days.

**Figure 14 materials-14-06813-f014:**
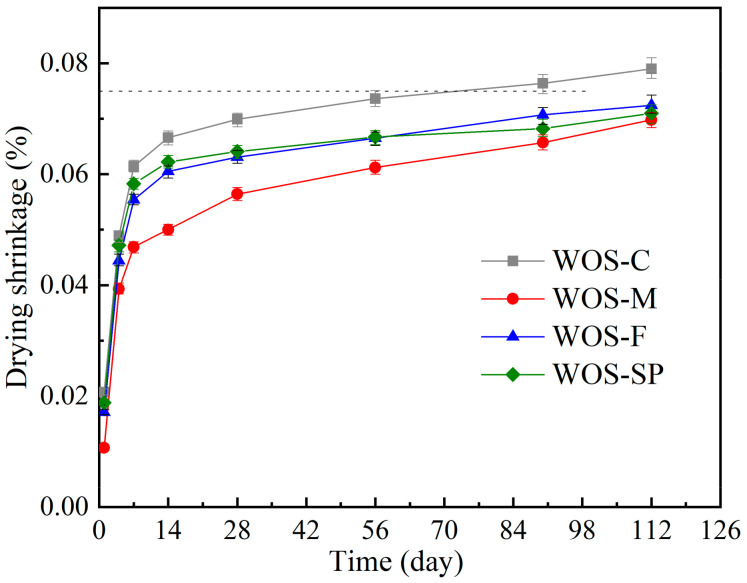
Effect of the particle size and water reducer on the drying shrinkage.

**Table 1 materials-14-06813-t001:** The chemical compositions and the mineral compositions of OPC.

Chemical Compositions (%)									Mineral Compositions (%)			
CaO	SiO_2_	Al_2_O_3_	Fe_2_O_3_	SO_3_	Na_2_O	K_2_O	MgO	TiO_2_	C_3_S	C_2_S	C_4_AF	C_3_A
60.16	21.35	4.94	2.71	1.96	1.00	0.48	0.46	0.15	60.74	16.18	14.17	6.66

**Table 2 materials-14-06813-t002:** Recipe for the test mortars.

Sample	Cement (kg/m^3^)	Fine Aggregate (kg/m^3^)				Global F.M.	SNF SP (kg/m^3^)	Water (kg/m^3^)	W/C
		River Sand	Fine WOS	Middle WOS	Coarse WOS				
WOS-C	608	1216	/	/	304	2.18	1.52 (0.25%)	273.6	0.45
WOS-M	608	1216	/	304	/	1.97	1.52 (0.25%)	273.6	0.45
WOS-F	608	1216	304	/	/	1.75	1.52 (0.25%)	273.6	0.45

## Data Availability

Data sharing is not applicable to this article.

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
