# Peer review of "Particle Size Effect of Oyster Shell on Mortar: Experimental Investigation and Modeling"

_materials, 2021, doi:10.3390/ma14226813_

Round 1
Reviewer 1 Report
A very interesting investigation. Here a few comments and suggestions:
1.) Could you please include a number how much WOS would be available compared to the demand of concrete? You mentioned the yearly production of concrete, but not the potential sum of WOS.
2.) Chapter 3.2.3: Could you please mention the R-spsquare value for the static elastic modulus.
3.) Chapter 3.2.4: Stress-strain curves: Coudl you please give an info, if the curves in Fig. 12 are single experiments or average of 3 experiments. Fig. 13: Please check the legend: It seams that WOS-C and WOS-F are exchanged.
4.) Chapter 3.2.5: Coudl you please check Fig. 14: Is WOS-SP = WOS-F2? If yes, please harmonize the name in Fig. and text. I recommend to use for WOS-C, WOS-M and WOS-F the same coulors as before, to guide the reader. Green colour could be used WOS-SP/F2.
5.) Chapter 3.3 Eco-efficiency. Coudl you please explain a bit more why the calcination of WOS is more energy-efficient compared with usage of river sand? Or do the WOS have puzzolanic effects which you didn't mention in the paper. In Chapter 2.1.2 the preoaration of WOS is not explained in great detail, may be you should explain a bit more.
6.) I have the feeling that the written English needs improvement, but I'm not a native speaker.
Author Response
Reviewer #1:
A very interesting investigation. Here a few comments and suggestions:
Thank you very much for your comments. The detailed clarifications and interpretations are provided below. Corresponding changes were marked in green in the manuscript. Please contact us if you have any additional comments for our paper.
1> Could you please include a number how much WOS would be available compared to the demand of concrete? You mentioned the yearly production of concrete, but not the potential sum of WOS.
Answer: Thank you for your suggestion. According to statistics in 2020, China's annual concrete output is about 2.9 billion cubic meters, of which coastal region needs about 800 million cubic meters of concrete, and fine aggregate occupies about 160 million cubic meters. We have added concrete output and potential sum of WOS, and more details in introduction.
2> Chapter 3.2.3: Could you please mention the R-spsquare value for the static elastic modulus.
Answer: R-spsquare value is 0.992. According to your suggestion, we have added R-spsquare value for the static elastic modulus in Fig.11.
3> Chapter 3.2.4: Stress-strain curves: Could you please give an info, if the curves in Fig. 12 are single experiments or average of 3 experiments. Fig. 13: Please check the legend: It seams that WOS-C and WOS-F are exchanged.
Answer: Thank you for your advice, we conducted tests on three test blocks, achieving three stress-strain curves. The middle line was selected for the final stress-strain curves of Fig. 12. For Fig.13, the legends of WOS-C and WOS-F got mixed up, and the error has been corrected. See the revised manuscript for corresponding revisions.
4> Chapter 3.2.5: Could you please check Fig. 14: Is WOS-SP = WOS-F2 ? If yes, please harmonize the name in Fig. and text. I recommend to use for WOS-C, WOS-M and WOS-F the same colours as before, to guide the reader. Green colour could be used WOS-SP/F2
Answer: We are sorry for we didn’t make it accurate. The WOS-SP and WOS-F2 are the same. We have harmonized the name in Fig. and text. According to your suggestion, we have modified the legend in Fig. 14 consistent with the other pictures.
5> Chapter 3.3 Eco-efficiency. Could you please explain a bit more why the calcination of WOS is more energy-efficient compared with usage of river sand ? Or do the WOS have puzzolanic effects which you didn't mention in the paper. In Chapter 2.1.2 the preparation of WOS is not explained in great detail, maybe you should explain a bit more.
Answer: We are sorry for the inaccuracy of the statement led you to misunderstanding. Actually, the crushed WOS without calcination was used as fine aggregate instead of river sand in this manuscript, instead of using calcined WOS to replace cement. Partial replacement of fine aggregate with crushed WOS does not involve energy-efficient. As a non-renewable resource, river sand is increasingly scarce. Replacing river sand with WOS could not only reduce the demand for river sand, but also properly dispose of waste WOS. We have modified the content on section 3.3.
The detail of crushed WOS preparation was as follows. WOS were first rinsed to remove contaminants. Then, they were dried at 105 °C in a drying oven for 5 h, crushed using a crushing machine. Finally, WOS with coarse, medium and fine particle sizes were obtained. We implement the details in section 2.1.2.
6> I have the feeling that the written English needs improvement, but I'm not a native speaker.
Answer: Thanks for your suggestion. The entire passage language has been polished. We have modified the grammatical problems and examined the format with reference to the requirements of Materials, more details in revised manuscript.

Reviewer 2 Report
The article presents research on the use of waste aggregate in the form of crushed shells in the production of cement mortars as a partial replacement for sand. Three different aggregate fractions in the form of shells were used. The influence of the use of this aggregate on selected basic parameters of mortars was investigated. There is no comparative recipe in the form of a mortar without WOS aggregate.
To improve the quality of the manuscript, the following corrections should be made:
- The WOS acronym should be explained both in the Introduction and in the Materials and Methods chapter. It is not enough that it is described in the abstract. The reader may not read the abstract.
- The title of section 2.5.3 should be "Water absorption and drying shrinkage".
- Figure 5: The methodology stated that the measurement of the setting time was performed on three samples. Therefore, error bars should be shown as standard deviation or confidence interval.
- The same applies to the results of compressive and flexural strength - the more so as a correlation (formula No. 5) has been demonstrated between the compressive strength and the flexural strength. Such dependencies should be described when performing more trials (more samples). The results may have been repeatable, but there are no error bars to prove it.
- Similarly as above - Fig. no. 11
- Are the average results from three measurements shown in Figures 12 and 13?
- The formatting of the bibliography should be adapted to the requirements of the journal.
Author Response
Reviewer #2:
The article presents research on the use of waste aggregate in the form of crushed shells in the production of cement mortars as a partial replacement for sand. Three different aggregate fractions in the form of shells were used. The influence of the use of this aggregate on selected basic parameters of mortars was investigated. There is no comparative recipe in the form of a mortar without WOS aggregate. To improve the quality of the manuscript, the following corrections should be made:
Thank you very much for your comments and suggestions for revision. In the previous study [1, 2], we studied the influence of different mixing ratios of crushed WOS (10%, 20%, 30%, 40%) on the properties of WOS mortar, and compared the differences between WOS mortar and ordinary mortar without WOS. It is found that 20% WOS is a suitable dosage. So a WOS content of 20% was selected. This paper focuses on the study of the influence of different particle sizes (fine particles, medium particles and coarse particles) on the properties of WOS mortar, so an ordinary mortar without WOS was not considered.
All the changes in the manuscript are highlighted in green. Detailed responses to these comments are listed point by point below. If you have any further questions or suggestions about our paper, please contact us again.
Reference
[1] Chen, D.; Zhang, P.C.; Pan, T.; Liao, Y.D.; Zhao, H. Evaluation of the eco-friendly crushed waste oyster shell mortars containing supplementary cementitious materials. J. Clean. Prod. 2019, 237, 117811.
[2] Chen, D.; Pan, T.; Yu, X.T.; Liao, Y.D.; Zhao, H. Properties of Hardened Mortars Containing Crushed Waste Oyster Shells. Environ. Eng. Sci. 2019, 36, 1079-1088.
1> The WOS acronym should be explained both in the Introduction and in the Materials and Methods chapter. It is not enough that it is described in the abstract. The reader may not read the abstract.
Answer: According to your suggestions, we have explained the WOS acronym in the Introduction and in the Materials and Methods chapter etc.
2> The title of section 2.5.3 should be “Water absorption and drying shrinkage”.
Answer: Thanks for your suggestion. The title of section 2.5.3 has been modified as “Water absorption and drying shrinkage”.
3> Figure 5: The methodology stated that the measurement of the setting time was performed on three samples. Therefore, error bars should be shown as standard deviation or confidence interval.
Answer: Thanks for your suggestion. The error bar for setting time in Fig. 5 has been added. In addition, we also carefully made similar modifications to Fig. 4, Fig. 6, Fig. 8, Fig. 10 and Fig. 14.
4> The same applies to the results of compressive and flexural strength - the more so as a correlation (formula No. 5) has been demonstrated between the compressive strength and the flexural strength. Such dependencies should be described when performing more trials (more samples). The results may have been repeatable, but there are no error bars to prove it.
Answer: Compressive strength and flexural strength have been measured by three specimens. In order to better describe the dependencies of the compressive strength and flexural strength, according to your suggestion, test results of all samples were listed in the figure to establish the relationship between compressive strength and flexural strength, and R2 was recalculated. The R2 was equal to 0.986. The modification is shown in Fig. 9.
5> Similarly as above - Fig. no. 11.
Answer: Thanks for your suggestion. Fig. 11 is similarly modified according to the modification method of Fig. 9. The R2 was equal to 0.992.
6> Are the average results from three measurements shown in Figures 12 and 13 ?
Answer: We conducted tests on three test blocks, achieving three stress-strain curves. The middle line was selected for the presentation stress-strain curves of Fig. 12 and Fig. 13.
7> The formatting of the bibliography should be adapted to the requirements of the journal.
Answer: Thanks for your suggestion, we have modified the formatting of the bibliography with reference to the requirements of Materials. In addition, the language of the full text has been revised, and more details in revised manuscript.

Reviewer 3 Report
General coments:
I am not a native English speaker but I suggest to revise English language because I found some mistakes in the manuscript. Better to include numbered lines for revisions.
Abstract:
The abstract introduction is confusing. The first phrase “different particle sizes” of what (WOS or aggregate or both).
….. dry shrinkage of WOS were systematically analysed… It should be mortars manufactured with WOS.
I think it should be included the particle sizes studied in the research.
Introduction section
I suggest to divide or join all the references and findings about the use of WOS in mortars, and other point or paragraph with all the references and findings about using WOS in concrete.
Materials section
I suggest to include more information about the WOS (cleaning, crushing, divided into different sizes…)
Table 1 include mineral composition quantities, I think it was determined by XRD. Mention the tests used for determining Chemical and mineral composition. I suggest a brief description about the findings of these tests.
I suggest to provide the standards used for determining the particle size, the absorption. I suggest to include the densities of the WOS in all fractions.
I suggest to include the long name of F.M.
Why the SNF SP was kept constant? I suggest to clarify it.
In the manuscript “Compressive strength and flexural strength were the average of three measurement”. Is the average of compressive strength six or three specimens?
Which is the standard used for static modulus? Could you give a brief description? How was the manufacture of these cylindrical samples? How was calculated the static modulus?
Results and discussion
I suggest to include the standard deviation in results showed in Figure 4,5, 6, 8 and 10.
Figure 6, the lines are not included in the legend. I suggest to include them.
I would like to know the density (apparent density) and the porosity of the mortars because maybe the results are also dependent of the density because fine aggregate fills the holes better than the coarse aggregates. There are many studies that studied the influence of the aggregate size in properties of mortars and I think they should be taken into account.
Which was the test to obtain the stress strain curves ?
I suggest to clarify better and discuss the drying shrinkage of mortars. New mortars were manufactured but they need to be better explained. How can be justified less shrinkage in finer aggregates?. Discuss and compare with other studies.
I suggest to compare the results obtained in WOS mortars with a control mortar or with other studies that analyse the aggregate size in mortars to see if the results are according with the scientific literature.
Author Response
Reviewer #3:
I am not a native English speaker but I suggest to revise English language because I found some mistakes in the manuscript. Better to include numbered lines for revisions.
Thanks for your suggestion. The manuscript has been checked and appropriate changes have been made in according to the your advice.
In response to your comments and suggestions, we have made corresponding changes in the manuscript and highlighted them in green. Detailed point-by-point changes is listed below. If you have any further questions or suggestions about our manuscript, please contact us again and we are happy to make further revisions to improve the manuscript.
1> Abstract:
â‘ The abstract introduction is confusing. The first phrase “different particle sizes” of what (WOS or aggregate or both).
Answer: Thanks for your suggestion. The first phrase “different particle sizes” means “different particle sizes of WOS”, we have modified it in the abstract.
② “… dry shrinkage of WOS were systematically analysed …” It should be mortars manufactured with WOS.
Answer: It refers to the dry shrinkage of WOS mortar, and we have modified it in the abstract.
③ I think it should be included the particle sizes studied in the research.
Answer: As your suggestion, we have supplemented the research on particle sizes in abstract.
2> Introduction section
I suggest to divide or join all the references and findings about the use of WOS in mortars, and other point or paragraph with all the references and findings about using WOS in concrete.
Answer: Thank you for suggestions. All the references and findings about the use of WOS in mortars or concrete have been divided, and more details can be seen in introduction.
3> Materials section
â‘ I suggest to include more information about the WOS (cleaning, crushing, divided into different sizes…)
Answer: Thank you for suggestions. The treating process (cleaning, crushing, divided into different sizes…) have been added to this manuscript, more details in section 2.1.2.
② Table 1 include mineral composition quantities, I think it was determined by XRD. Mention the tests used for determining Chemical and mineral composition. I suggest a brief description about the findings of these tests.
Answer: We are sorry for confusing. The chemical and mineral composition of cement are provided by the factory. In addition, the performance indexes of cement meet the standard of GB175 (2007) requirements. The modification is shown in section 2.1.1.
③ I suggest to provide the standards used for determining the particle size, the absorption. I suggest to include the densities of the WOS in all fractions.
Answer: As you suggested, we have provided the standards used for determining the particle size, and the absorption has been added as follows ASTM C1585 (2013). The densities of the coarse, medium, and fine crushed WOS particle is 1284 kg/m3, 1299 kg/m3 and 1354 kg/m3, respectively. The test results of related densities have been supplemented in section 2.1.2.
â‘Ł I suggest to include the long name of F.M.
Answer: The long name of F.M. is fineness modulus. As you suggested, we have complemented F.M.'s long name both in the Introduction and in the Materials and Methods chapter etc.
⑤ Why the SNF SP was kept constant? I suggest to clarify it.
Answer: According to the pre-test results, the amount of water reducing agent on the premise of meeting the workability requirements for each WOS mortar was different, but the amount was all close to 0.25%. In order to compare the properties of WOS mortars with different particle sizes, the same amount of SNF SP (0.25%) was used. Relevant explanations have been supplemented in section 2.1.3.
â‘Ą In the manuscript “Compressive strength and flexural strength were the average of three measurement”. Is the average of compressive strength six or three specimens?
Answer: We are sorry for confusing. The compressive strength test is average of 3 specimens. The related modification is shown in section 2.5.1.
⑦ Which is the standard used for static modulus? Could you give a brief description? How was the manufacture of these cylindrical samples? How was calculated the static modulus?
Answer: The static elasticity modulus is tested according to standard of EN-197-1-211. According to the specification, the cylinder test block is made of steel mold. The test method of static modulus is as follows: The relationship between strain and stress is obtained through equal strain loading method. And then taking two points in the elastic segment to calculate the slope, the static elasticity modulus is achieved in the end. The relevant content of static elasticity modulus has been added, and more details can be seen in section 2.5.2.
4> Results and discussion
â‘ I suggest to include the standard deviation in results showed in Figure 4,5, 6, 8 and 10.
Answer: Thanks for your suggestion. The standard deviation in Fig. 4, 5, 6, 8 and 10 has been added. Additionally, we also added the standard deviation in Fig. 14.
② Figure 6, the lines are not included in the legend. I suggest to include them.
Answer: The lines of legend in Fig. 6 have been added, more details in Fig. 6.
③ I would like to know the density (apparent density) and the porosity of the mortars because maybe the results are also dependent of the density because fine aggregate fills the holes better than the coarse aggregates. There are many studies that studied the influence of the aggregate size in properties of mortars and I think they should be taken into account.
Answer: You are right. Many studies show that the density and porosity of the WOS are both important indicators to evaluate the properties of mortar. And both of density and porosity can be used to reflect the microstructure denseness of the material. In this paper, the porosity of WOS mortar was tested. The porosity of WOS mortar with different particle sizes (WOS-F, WOS-M and WOS-C) is 8.74%, 10.26% and 10.48%, respectively. The test results of related porosity have been supplemented in section 2.2.
At the same time, according to your suggestions, we supplement and cite other scholars’ studies on the impact of aggregate particle size on the compressive strength, flexural strength, static elasticity modulus, dry shrinkage rate, stress-strain curve and other indicators of mortar in the result analysis to further verify the conclusions of this paper.
â‘Ł Which was the test to obtain the stress strain curves ?
Answer: Thank you for suggestions. Three cylinder samples of φ 50 mm × 100 mm were used for the stress-strain test. Before the stress-strain of the sample was measured, a linear variable displacement transducer (LVDT) was fixed on two opposite sides of the sample. The test of uni-axial compression was carried out at constant strain rate of 0.001 mm/s using the auto-compensated and auto-equilibrated triaxial cell system. The axial stress and the axial strain were collected by the computer, related details in section 2.5.4.
⑤ I suggest to clarify better and discuss the drying shrinkage of mortars. New mortars were manufactured but they need to be better explained. How can be justified less shrinkage in finer aggregates? Discuss and compare with other studies.
Answer: Thanks for your suggestions. Dry shrinkage refers to that the mortar in the air, as a result of moisture diffusion to the external environment, can cause the phenomenon of volume contraction. The related contents have been supplemented in section 3.2.5, and more details are as follows:
The dry shrinkage of mortar is mainly determined by the capillary tension in mortar. Adding WOS can effectively improve particle size distribution and refine pore structure in mortar. In addition, the finer the pore structure of mortar is, the smaller the internal capillary tension will be. With the same amount of WOS, the smaller the aggregate particle size is, the smaller the pore size of mortar is, the more refined the pore structure of mortar will. Therefore, the smaller the WOS particle size is, the smaller the dry shrinkage value will be. Relevant references have been supplemented in the manuscript, see references for details.
â‘Ą I suggest to compare the results obtained in WOS mortars with a control mortar or with other studies that analysed the aggregate size in mortars to see if the results are according with the scientific literature.
Answer: Thank you for suggestions. According to your suggestions, the compressive strength, flexural strength, elastic modulus, stress-strain curve and other performance indexes are compared and analyzed with other literatures to fully verify the scientific nature of this study furtherly. The related contents have been supplemented in section 3.2.1 ~ 3.2.5.

Round 2
Reviewer 3 Report
I think that the manuscript has been improved from the first version. But I still think that it should be checked by a native English because I read sentences that have mistaken.
I suggest to rewrite the abstract according the journal requirements: it may content introduction, methodology, results and the main conclusions. I suggest to highlight the findings. Do not include abbreviations without description (WOS-SP) and I suggest to make it more general.
The sentence “Aggregates used here were in saturated surface dry condition” should be described. Dry with a towel? Vibration and non-absorbent towel? I suggest to include a brief description.
Author Response
1> I think that the manuscript has been improved from the first version. But I still think that it should be checked by a native English because I read sentences that have mistaken.
Answer: Thank you for your accepting. In addition, according to your suggestions, we have made our manuscript checked by a native English-speaking colleague. The entire grammar of the manuscript has been revised and polished again. Corresponding revisions made to the manuscript have been marked up using the “Track Changes” function, and more details in revised manuscript.
2> I suggest to rewrite the abstract according the journal requirements: it may content introduction, methodology, results and the main conclusions. I suggest to highlight the findings. Do not include abbreviations without description (WOS-SP) and I suggest to make it more general.
Answer: Thank you for your advice. We have rewritten the abstract according to the journal requirements. The introduction, methodology, results and the main conclusions have been added in the abstract. Meanwhile, the findings have been highlighted in the conclusion of the abstract. Some abbreviations (WOS-SP et al.) have been described in detail. Relevant modification can be seen in the abstract.
3> The sentence “Aggregates used here were in saturated surface dry condition” should be described. Dry with a towel? Vibration and non-absorbent towel? I suggest to include a brief description.
Answer: Thank you for your suggestion. For the “aggregates used here were in saturated surface dry condition”, our method is as follow: First, the aggregate is soaked in water to reach a certain humidity. Then, the aggregate was taken out and dried naturally in the laboratory until there was no obvious moisture on the aggregate surface. All the internal pores of aggregate are saturated with water absorption, but there is no obvious water on the surface. Relevant modification can be seen in the section 2.3.

This manuscript is a resubmission of an earlier submission. The following is a list of the peer review reports and author responses from that submission.